# Urine Flow Cytometry and Dipstick Analysis in Diagnosing Bacteriuria and Urinary Tract Infections among Adults in the Emergency Department—A Diagnostic Accuracy Trial

**DOI:** 10.3390/diagnostics14040412

**Published:** 2024-02-13

**Authors:** Mathias Amdi Hertz, Isik Somuncu Johansen, Flemming S. Rosenvinge, Claus Lohman Brasen, Eline Sandvig Andersen, Claus Østergaard, Thor Aage Skovsted, Eva Rabing Brix Petersen, Stig Lønberg Nielsen, Christian Backer Mogensen, Helene Skjøt-Arkil

**Affiliations:** 1Department of Infectious Diseases, Odense University Hospital, 5000 Odense, Denmark; isik.somuncu.johansen@rsyd.dk (I.S.J.);; 2Research Unit of Infectious Diseases, Department of Clinical Research, Faculty of Health Sciences, University of Southern Denmark, 5000 Odense, Denmark; 3Department of Clinical Microbiology, Odense University Hospital, 5000 Odense, Denmark; 4Research Unit of Clinical Microbiology, University of Southern Denmark, 5000 Odense, Denmark; 5Department of Regional Health Research, Faculty of Health Sciences, University of Southern Denmark, 5000 Odense, Denmarkcbm1@rsyd.dk (C.B.M.); hsa@rsyd.dk (H.S.-A.); 6Department of Biochemistry and Immunology, Lillebælt Hospital—University Hospital of Southern Denmark, 6000 Kolding, Denmark; 7Department of Clinical Microbiology, Lillebælt Hospital—University Hospital of Southern Denmark, 7100 Vejle, Denmark; 8Department of Biochemistry and Immunology, University Hospital of Southern Denmark, 6200 Aabenraa, Denmark; 9Emergency Department, University Hospital of Southern Denmark, 6200 Aabenraa, Denmark

**Keywords:** urinary tract infection, urine flow cytometry, urine culture, urine dipstick analysis, bacteriuria, urinalysis

## Abstract

Urinary tract infections (UTIs) are a leading infectious cause of emergency department admission. Early UTI diagnosis is challenging, and a faster, preferably point-of-care urine analysis is necessary. We aimed to evaluate the diagnostic accuracy of urine flow cytometry (UFC) and urine dipstick analysis (UDA) in identifying bacteriuria and UTIs. This study included adults suspected of an infection admitted to three Danish emergency departments. UFC and UDA were the index tests, and urine culture and an expert panel diagnosis were the reference tests. We used logistic regression and receiver operator characteristics curves to find each test’s optimal model and cut-off. We enrolled 966 patients and performed urine cultures on 786. Urine culture was positive in 337, and 200 patients were diagnosed with a UTI. The UFC model ruled out bacteriuria in 10.9% with a negative predictive value (NPV) of 94.6% and ruled out UTI in 38.6% with an NPV of 97.0%. UDA ruled out bacteriuria in 52.1% with an NPV of 79.2% and UTI in 52.8% with an NPV of 93.9%. Neither UFC nor UDA performed well in ruling out bacteriuria in our population. In contrast, both tests ruled out UTI safely and in clinically relevant numbers.

## 1. Introduction

### 1.1. Background

Urinary tract infections (UTIs) are the second most common infection presenting in emergency departments (EDs) [1]. UTIs cause over 2 million ED visits annually in the United States, and in Denmark, over 15,000 patients are admitted yearly with a UTI [2,3]. Urine cultures are essential for the etiological diagnosis and choice of antibiotic treatment. Consequently, the threshold for performing urine cultures is low in EDs, leading to many unnecessary urine cultures [4].

Although urine cultures are highly sensitive and specific for bacteriuria, their performance in diagnosing UTIs suffers. Sensitivity can be as low as 45%, specificity as low as 72%, and negative predictive value (NPV) as low as 50% [5,6,7].

Despite this, bacteriuria (positive urine culture) is repeatedly equated to a diagnosis of UTI in the literature, though often supported by the presence of symptoms [8,9]. Furthermore, the result of a urine culture is only available after 24 to 48 h, which makes it unusable for emergency diagnostics.

Symptoms of UTI can be variable and nonspecific, particularly in the elderly [10]. Early goal-directed treatment of infected patients in the ED necessitates fast and accurate urine analysis. Furthermore, better diagnostic capabilities can reduce antimicrobial resistance by improving antibiotic prescription in the ED [11].

Urine flow cytometry (UFC) counts and differentiates cells in urine by staining them with fluorescent markers, illuminating them with a laser, and recording the scattered light [12,13,14]. Though not a point-of-care analysis, UFC remains relevant as a rule-out test when antibiotic treatment can be delayed until initial lab results are available [15]. White blood cell counts and bacterial counts have been investigated for bacteriuria and UTI. Other cell types, like squamous epithelial cells, have been evaluated as possible indicators of contamination, but with mixed results [16]. Most studies have evaluated UFC using urine culture as the gold standard, with varying results depending on the urine culture and UFC cut-offs [16,17,18,19].

Urine dipstick analysis (UDA) is a fast, inexpensive, and easy point-of-care test that detects leukocyte esterase, indicating inflammation and nitrite indicating nitrite-producing bacteria. Though studies on the diagnostic accuracy of UDA to diagnose UTIs exist, to our knowledge, only one study has been conducted in which a positive urine culture was not required for a definite diagnosis of UTI [20].

### 1.2. Aim and Objectives

The aim of the study was to investigate if UFC and UDA can be used to exclude bacteriuria and diagnose UTI in an ED setting.

Our primary objective was to determine the diagnostic accuracy of UFC and UDA for detecting bacteriuria and to calculate the optimal cut-offs.

The secondary objective was to determine UFC and UDA’s diagnostic accuracy and cut-offs for detecting UTI.

## 2. Materials and Methods

### 2.1. Study Design

This study was a multicentre type 1 diagnostic accuracy trial with prospective data collection. It presents one of the objectives of the multifaceted INDEED (Infectious Diseases in Emergency Departments) study, which investigates new diagnostic tools and working methods that support rapid and accurate diagnosis to avoid unnecessary antibiotic prescriptions in EDs [11].

The study was designed and reported in accordance with the STARD guidelines [21].

### 2.2. Participants

We enrolled patients from the medical ED of three hospitals in Southern Denmark: University Hospital of Southern Denmark in Aabenraa and Sønderborg, Lillebælt Hospital in Kolding, and Odense University Hospital in Odense. From March 2021 until February 2022, patients were enrolled consecutively on weekdays during the daytime and evenings. The three hospitals are part of Denmark’s publicly funded healthcare system and serve a defined catchment population of 775,000, comprising both rural and urban populations.

A staff of six healthcare-educated project assistants screened all new patients admitted to the ED for an indication of suspected infection or unspecific complaints. After the initial ED assessment, the project assistant asked the attending physician whether an infection was suspected. If an infection was suspected, the patient was screened for eligibility and invited to participate.

Adults aged 18 years or older who were admitted and able to give informed consent were invited unless they met one or more of the following exclusion criteria: If participation would delay life-saving treatment, prior admission (>24 h) within the last 14 days before the current admission, verified COVID-19 within 14 days before admission, pregnant patients, or patients with severe immunodeficiency.

### 2.3. Tests and Variables

#### 2.3.1. Urine Sample Logistics

The project assistants or trained nurses collected the urine samples. As part of the standard of care, urine samples are collected from all patients suspected of having an infection. If a urine sample had already been collected by a nurse and sent for culture but not to UFC and UDA, the project assistant collected an additional sample to ensure the same material was analyzed in all three tests. The urine collection method and sampling time were recorded in the data collection tool.

#### 2.3.2. Urine Flow Cytometry—Index Test

UFC analysis was performed by the Sysmex UF-5000 (Sysmex Corporation, Kobe, Japan) with an automated rinse (settings 0,1,1,7,7) after samples with a high bacterial count to avoid carryover [22]. Analysis results of bacterial count per µL (BACT/µL), WBC count per µL (WBC/µL), squamous epithelial cell count per µL, and analysis time were recorded from the Sysmex UF-5000. Analysis was performed by either the project assistants or lab workers trained to use the Sysmex UF-5000. Only results with a time from urine collection to analysis between zero and two hours were used for our calculations. The urine culture results and the expert panel diagnosis were not available at the time of analysis.

#### 2.3.3. Urine Dipstick Analysis—Index Test

We used Siemens Multistix 7 (Siemens Healthcare GmbH, Erlangen, Germany) dipsticks for analyses, which were either automatically read by Siemens Clinitek status + analyser (Siemens Healthcare GmbH, Erlangen, Germany) or manually read in a few cases. The project assistant recorded leukocyte esterase and nitrite in the data collection tool. UDA was performed after urine was sampled for UFC to avoid pigments in the urine. The urine culture results and the expert panel diagnosis were not available at the time of analysis.

#### 2.3.4. Urine Cultures—Reference Test

Urine samples were cultured in one of three clinical microbiology departments at the study sites. Different transport methods, culture methods, and cut-offs and definitions of bacteriuria were applied, as can be seen in detail in the Appendix B. UFC and UDA results were unavailable for lab workers performing the urine cultures.

#### 2.3.5. Urinary Tract Infection Diagnosis—Reference Test

An expert panel retrospectively assigned the clinical UTI diagnosis based on available information in the medical record within the first week after enrolment. At each site, a team of one emergency medicine specialist and one infectious diseases specialist independently reviewed the medical records. A consensus agreement was reached in case of discordant diagnosis. The experts could not be blinded to urinary dipstick analysis results but were blinded to the WBC count in urine flow cytometry. In one hospital, it was not possible to blind the experts to the BACT count, but this parameter was not used for the final calculations.

#### 2.3.6. Other Variables

After obtaining consent, the project assistants collected information on age, sex, and type of urinary catheter on admission. All information was recorded in the online data collection tool (REDCap version 10.8.3 to version 12.2.1 by Vanderbilt University, Nashville, TN, USA).

### 2.4. Statistical Analysis

Patients’ characteristics were summarized using descriptive statistics. Categorical variables were reported as proportions, while continuous variables were reported as median and interquartile ranges.

Univariate and multivariate logistic regression analyses and the area under the receiving operator characteristics curve (AUROC) were used to find the best model for cut-off analysis. We analyzed the two UFC (WBC/µL and BACT/µL) parameters separately and in combination and urine dipstick (leukocytes and nitrite) results individually and in combination. Since squamous epithelial cells could indicate contamination, we also tested a model considering the interaction between bacterial count and squamous epithelial cells.

We selected the model with the highest AUROC for each urine analysis method (UFC and UDA) and outcome (bacteriuria and UTI).

Once the best model was selected, we chose a high-sensitivity cut-off for the bacteriuria analysis to safely rule out the need for urine culture. We established three cut-offs for analyzing UTI, optimized for sensitivity, specificity, and diagnostic accuracy (DA) to reflect a clinician’s different interests in test accuracy. We set a goal of 95% for the sensitivity and specificity cut-offs; for DA cut-offs, we chose the highest. Confidence intervals for sensitivity, specificity, and DA were calculated using Clopper–Pearson confidence intervals, and predictive values were the standard logit confidence intervals.

For bacteriuria and UTI, we performed subgroup analyses on sex assigned at birth since prior studies found a difference between the sexes [23,24]. For bacteriuria, we performed a sensitivity analysis for the inclusion site due to the difference in culture methods. For UTI, we recalculated and compared the AUROC without the non-infected patients because including them could lead to bias if the tests were sensitive to infections in general.

All statistical analyses were performed using Stata (StataCorp. 2023. Stata Statistical Software: Release 18. College Station, TX, USA: StataCorp LLC).

## 3. Results

### 3.1. Participants

Of 2.197 patients assessed for eligibility, 1231 were excluded or declined to participate. We enrolled 966 patients in the study and performed urine cultures on 786 (81.4%), of which 337 (42.9%) were positive. Out of 516 patients where UFC was performed, 512 also had urine cultures performed, while out of 812 patients who had UDA performed, 776 had urine cultures performed. Of the enrolled patients, 200 (20.7%) were diagnosed with UTI, 589 (61.0%) had other infections, and 177 (18.3%) had no infection (Figure 1).

Baseline characteristics are summarized in Table 1. The patients with bacteriuria (*n* = 337) had a median age of 76 (IQR 16); 180 (53.4%) were male, 62 (18.4%) had a urinary catheter, and 147 (43.6%) had a UTI. The patients with UTI (*n* = 200) had a median age of 76 (IQR 17), 116 (58.0%) were male, and 47 (23.5%) had a urinary catheter. Urine culture was performed in 185 patients (92.5%), of whom 147 (79.5%) had bacteriuria.

### 3.2. Diagnostic Accuracy of UFC and UDA for Bacteriuria

Table 2 summarizes the diagnostic performance of UFC and UDA to detect bacteriuria. We found that a UFC model using BACT/µL and WBC/µL had an AUROC of 0.84 (95% CI 0.80–0.87). Models including an interaction between BACT/µL and squamous epithelial cell count/µL had a lower AUROC than those without. We calculated the 95% sensitivity cut-off for BACT to be 7/µL and 3.2/µL for WBC. In the model, if either variable was above the cut-off, we calculated a sensitivity of 98.6% (95% CI 95.9–99.7), a specificity of 17.7% (95% CI 13.6–22.5), a positive predictive value (PPV) of 46.0% (95% CI 44.7–47.4), an NPV of 94.6% (95% CI 84.8–98.2), and a DA of 51.4% (95% CI 46.9–55.8).

The best model for UDA used leukocyte esterase and nitrite and had an AUROC of 0.78 (95% CI 0.76–0.82). We calculated the parameters for all the combinations of cut-offs, including both variables, and found that a cut-off of either positive leukocytes +1 or higher or nitrite had the highest sensitivity of 74.5% (95% CI 69.5–79.2) and NPV of 79.2% (95% CI 75.8–82.2).

### 3.3. Diagnostic Accuracy of UFC and UDA for Urinary Tract Infection

Table 2 summarizes the diagnostic performance of UFC and UDA to detect UTI. The best UFC model for diagnosing UTI was WBC/µL, with an AUROC of 0.86 (95% CI 0.82–0.89). We found no added value of combining with bacteria per µL and squamous epithelial cells per µL.

We identified a cut-off of 15 WBC/µL for the 95% sensitivity target (specificity 48.7% (95% CI 43.7–53.8), PPV 35.4% (95% CI 33.1–37.9), NPV 97.0% (95% CI 93.6–98.6), and DA 59.3% (95% CI 54.9–63.6) (Table 2).

A cut-off of 1125 WBC/µL was identified for the 95% specificity target (sensitivity 33.9% (95% CI 25.4–43.2), PPV 67.8% (95% CI 55.9–77.7), NPV 82.9% (95% CI 81.0–84.7), and DA of 81.2% (95% CI 77.6–84.5). The highest DA was found with a cut-off of 448 WBC/µL (sensitivity 50.9% (95% CI 41.5–60.2), specificity 91.5% (95% CI 88.3 to 94.0), PPV 63.8% (95% CI 55.0–71.8), NPV 86.3% (95 CI 83.9–88.3), and DA of 82.2% (95% CI 78.6–85.4)).

For UDA, a model using both leukocytes and nitrite had the highest AUROC for diagnosing UTI (0.81, 95% CI 0.77–0.84). The closest cut-off to 95% sensitivity was either leukocytes +1 or higher or nitrite-positive, giving an NPV of 93.9% (95%CI 91.5–95.7%) in our population. The closest cut-off to 95% specificity was leukocytes +2 or higher and positive nitrite, while a cut-off of leukocytes +3 or higher and positive nitrite had the highest DA (Table 2).

### 3.4. Additional Analyses

Subgroup analysis of UFC to rule out bacteriuria found a difference between the sexes. The AUROC was 0.90 (95% CI 0.86–0.94) for men and 0.74 (95% CI 0.68–0.81) for women.

In UDA, we also found a difference: AUROC was 0.83 (95% CI 0.79–0.87) in males, while it was 0.73 (95% CI 0.68–0.78) in women when testing for bacteriuria.

The same difference was found between the sexes for diagnosing UTI. For UFC, the AUROC was 0.89 (95% CI 0.85–0.93) for men but 0.80 (95% CI 0.73–0.87) for women. For UDA, the AUROC was 0.83 (95% CI 0.78–0.87) for men but 0.74 (95% CI 0.68–0.81) for women (Table 3).

Sensitivity analysis of the three sites showed an AUROC of 0.81 (95% CI 0.75–0.86) for Lillebælt Hospital, 0.88 (95% CI 0.82–0.94) for Odense University Hospital, and 0.85 (95% CI 0.78–0.92) for University Hospital of Southern Denmark for UFC to rule out bacteriuria, respectively.

When analyzing for UTI, excluding individuals with no infection resulted in a slight change in the AUROC. The values were 0.86 (95% CI 0.82–0.90) for UFC and 0.80 (95% CI 0.77–0.84) for UDA.

### 3.5. Adverse Events from Performing the Tests

Urine cultures and UFC and UDA testing were performed on urine samples taken as part of the standard of care. These tests are safe and cannot cause any adverse events. Since the expert panel’s assessment was conducted retrospectively and had no direct impact on the patient, no adverse events could have occurred.

## 4. Discussion

### 4.1. Key Results

In this prospective trial to evaluate the diagnostic accuracy of UFC and UDA to exclude bacteriuria and to diagnose UTI in the ED, we found that the best models for excluding bacteriuria were a combination of BACT/µL and WBC/µL for UFC (AUROC 0.83) and leukocytes and nitrite for UDA (AUROC 0.79).

We calculated the cut-offs of WBC/µL and BACT/µL for the 95% sensitivity target to be 7 per µL and 3.2 per µL, respectively. When we combined these into one test and only required one to be above the cut-off, we found a sensitivity of 98.6%, a specificity of 17.7%, a PPV of 46.1%, an NPV of 94.6%, and a DA of 51.4% for detecting bacteriuria.

UDA, with a cut-off of detecting either leukocytes or nitrite, provided a sensitivity of 74.5%, an NPV of 79.2%, and a DA of 72.9% for bacteriuria.

The best models for diagnosing UTI were WBC/µL alone for UFC and a combination of leukocytes and nitrite for UDA. A WBC/µL cut-off of 15/µL provided a sensitivity of 94.9%, an NPV of 97.0%, and a DA of 59.3%. A UDA cut-off of either leukocytes or nitrite being positive provided a sensitivity of 86.2%, an NPV of 93.9%, and a DA of 69.7% for diagnosing UTI.

### 4.2. Study Limitations

Our pragmatic study had some limitations. First, it was not possible to have urine cultures performed the same way with the same cut-offs and definitions at the three sites. Different culture methods at the three sites could have resulted in variations in the diagnostic parameters and AUROCs of the subgroup analysis. The cut-offs reflect this heterogeneity and can be used in settings with varying cut-offs and culture methods.

Urine samples for culture and UDA were obtained from most patients, but only half of the included patients had a UFC analysis performed. Patients who are anuric due to sepsis or urinary tract obstruction cannot provide a urine sample and could be argued to be more likely to have bacteriuria or UTI. Therefore, we must suspect a selection bias in the missing urine samples, and we classified them as “missing not at random”. Since these patients would have more severe disease, we argue they would improve the test performance if they were possible to test.

Samples collected but not analyzed were due to insufficient urine volume, lost samples, missing results, or a time to UFC analysis exceeding 2 h. We consider these to be “missing completely at random”, as the possible diagnosis or severity of the disease did not influence them.

Since this study is part of a more extensive study, it was impossible to blind the expert panel to the results of UDA, as the results were available in the patient’s charts. This could overestimate the performance of UDA.

As we would expect some influence from UDA on the expert panel diagnosis but no influence from UFC, we could not perform a comparison or net reclassification improvement of the two analyses.

### 4.3. Implications for Practice

Our findings on UFC and bacteriuria align with other studies. Comparable studies report an AUROC of 0.78–0.94 [8,16,17,25], which varies with the definition of bacteriuria and the variables selected for the different models. While most studies in line with our results found BACT/µL and WBC/µL to be the optimal model [8,25,26,27], others found the best model to be BACT/µL alone [17,18,28] or added red blood cells or squamous epithelial cells to the model for higher AUROCs or detection of contamination [15,16].

Our calculated cut-offs for BACT/µL and WBC/µL correspond reasonably with the literature when considering our target of a high sensitivity [8,18,19,27]. Studies aiming for other targets generally found higher cut-off values [8,9]. In our population using our model combining WBC/µL and BACT/µL, cut-off values of 7 per µL and 3.2 per µL, respectively, and a test where one or the other is positive, 10.9% of urine cultures could be omitted, leading to 10 UFC tests needing to be performed to avoid one urine culture. Among the omitted tests, 5.3% were positive, indicating that UFC is safe for ruling out bacteriuria, though in numbers too small to be clinically valuable (Table 2).

Our AUROCs for UDA for detecting bacteriuria are similar to prior studies, with some variation as expected from the different definitions of bacteriuria [8,16]. Our sensitivity and specificity findings align with previous studies, while PPV and NPV differ, reflecting different populations [8,16,28,29]. Our model for UDA using a positive result for either leukocytes or nitrite as cut-off would have reduced urine cultures by 52.1% in our population. Of these, 20.8% were positive, making the test unfeasible for clinical use (Table 2).

Urine cultures are invaluable in determining the etiology and resistance patterns of UTIs and thus provide vital information for the treatment of UTIs. However, poor diagnostic performance for UTIs and the fact that they take 24 to 48 h makes them unsuitable for diagnosing UTIs in the ED. Thus, faster and better diagnostic tools are needed in the ED.

It can be challenging to compare the outcomes of our diagnostic models for UTI because most studies consider a positive urine culture as equivalent to a UTI diagnosis. This assumption has led to a lack of studies that do not require a positive urine culture to confirm a definitive UTI diagnosis.

Using UFC to rule out UTI, our AUROC of 0.86 was almost equal to the only study with no requirement of positive urine culture and similar clinical diagnosis [20]. The same study set its sensitivity target to 90% and thus had a higher WBC/µL cut-off. With our cut-off of 15 WBC/µL, a negative test will rule out a UTI with 97.0% certainty, which in our population would rule out a UTI in 200 patients while only missing six actual UTIs, making it a very safe and usable test for ruling out UTI in infected patients. Similarly, a negative UDA test for both nitrite and leukocytes will rule out a UTI with 93.9% certainty, ruling out UTI in 426 patients in our population; of these, though, 26 had a UTI. Considering those numbers, UDA remains a fast, simple, inexpensive, and reasonably safe test for ruling out UTIs in patients with an infection.

Conversely, our results showed that using UFC or UDA, even with a cut-off optimized for high specificity, to rule in UTIs is unsafe due to many false positives and cannot be recommended.

## 5. Conclusions

The results of this study indicate that the advantage of using UFC or UDA to reduce unnecessary urine cultures is minimal. However, in the ED, where fast results can reduce the use of empirical broad-spectrum antibiotics and the amount of further diagnostics needed, UFC and UDA can be important tools for ruling out UTIs until better tests are available.

## Figures and Tables

**Figure 1 diagnostics-14-00412-f001:**
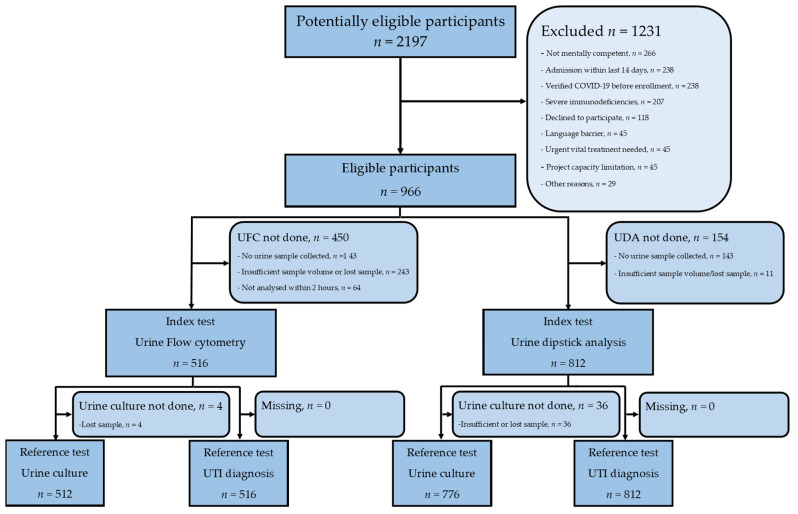
Flowchart of included patients and tests. UFC, Urine flow cytometry; UDA, urine dipstick analysis.

**Table 1 diagnostics-14-00412-t001:** Characteristics of patients admitted to the emergency department with suspicion of infection stratified by bacteriuria and expert panel diagnosis.

Patient Characteristics		Urine Culture		Expert Panel Diagnosis	
*n* = 966 unless stated otherwise.		*n* = 786		*n* = 966	
		Bacteriuria	No Bacteriuria	Urinary tract infection	No urinary tract infection
no. (%)		337 (42.9%)	449 (57.1)	200 (20.7)	766 (79.3)
Age, years, median (IQR)		76 (17)	69 (27)	76 (17)	72 (23)
Sex, no. (%)	Male	180 (53.4)	240 (53.4)	116 (58)	405 (52.9)
Urine sample method, *n* = 786 (UC)/822 (EPD), no. (%)	Midstream	181 (53.7)	349 (77.7)	105 (55.6)	444 (70.8)
	Catheter	53 (15.7)	23 (5.1)	34 (18.0)	47 (7.5)
	Sterile intermittent catheterization	36 (10.7)	17 (3.8)	23 (12.2)	34 (5.4)
	Bedpan or urine bottle	60 (17.8)	55 (12.3)	23 (12.2)	99 (15.8)
	Other/unknown	7 (2.1)	5 (1.1)	7 (3.7)	6 (1.0)
Catheter type before admission no. (%)	None	275 (81.6)	434 (96.7)	153 (76.5)	727 (94.9)
(more than one catheter type possible)	Catheter a demeure	34 (10.1)	5 (1.1)	24 (12)	20 (2.6)
	Clean intermittent catheterization	7 (2.1)	7 (1.6)	9 (4.5)	6 (0.8)
	Sterile intermittent catheterization	3 (0.9)	1 (0.2)	2 (1)	2 (0.3)
	JJ catheter, nephrostomy catheter, suprapubic, or urostomy catheter	20 (5.9)	2 (0.4)	14 (7)	11 (1.4)
Antibiotic treatment before urine culture, *n* = 786, no. (%)	Yes	118 (35.0)	175 (39.0)	82 (44.3)	211 (35.1)
Urine culture, *n* = 786, no. (%)	Positive	337 (100)	0 (0)	147 (79.5)	190 (31.6)
Expert panel diagnosis of UTI, no. (%)	Yes	147 (43.6)	38 (8.5)	200 (100)	0 (0)
Urine dipstick analysis					
Urine Leukocytes, *n* = 776 (UC)/812 (EPD), no. (%)	Negative	95 (28.8)	326 (73.1)	29 (15.3)	326 (67.1)
	+	57 (17.3)	54 (12.1)	32 (16.9)	81 (13.0)
	++	86 (26.1)	46 (10.3)	62 (32.8)	75 (12.0)
	+++	44 (13.3)	12 (2.7)	30 (15.9)	29 (4.9)
	++++	48 (14.6)	8 (1.8)	36 (19.0)	20 (3.2)
Urine nitrite, *n* = 776 (UC)/812 (EPD), no. (%)	Positive	100 (30.3)	11 (2.5)	61 (32.3)	56 (9.0)
Urine flow cytometry					
White blood cells/µL, *n* = 512 (UC)/516 (EPD), median (IQR)		205 (1078)	11.8 (44.4)	485 (2321)	17 (70)
Bacterial cells/µL, *n* = 512 (UC)/516 (EPD), median (IQR)		1438 (15,483)	24.9 (190)	1801 (22,736)	50 (535)
Squamous epithelial cells/µL, *n* = 512 (UC)/516 (EPD), median (IQR)		4.2 (10.2)	3.4 (15.2)	4.1 (8.1)	3.9 (14.4)

EPD, Expert panel diagnosis; UC, urine culture.

**Table 2 diagnostics-14-00412-t002:** Model AUROC, diagnostic values, and cross tabulations for each test stratified by reference test.

Reference Test	Index Test	Variables and Cut-Off	Model AUROC (95%Ci)	Sensitivity	Specificity	PPV	NPV	DA	TP	FP	FN	TN
Bacteriuria	UFC	BACT/µL–7/µL	0.809 (0.772–0.847)	95.3%	29.1%	48.9%	89.7%	56.6%	203	212	10	87
*n* = 786	*n* = 512	WBC/µL–3.2/µL	0.800 (0.761–0.839)	95.8%	25.4%	47.8%	89.4%	54.7%	204	223	9	76
		BACT/µL–7/µL or WBC/µL–3.2/µL	0.835 (0.800–0.870)	98.6%	17.7%	46.1%	94.6%	51.4%	210	246	3	53
		BACT/µL–7/µL and WBC/µL–3.2/µL	-	92.5%	36.8%	51.0%	87.3%	60.0%	197	189	16	110
	UDA	Leucocytes +1	0.751 (0.718–0.783)	71.2%	73.1%	66.2%	77.4%	72.3%	235	120	95	326
	*n* = 776	Leucocytes +2	-	53.9%	85.2%	73.0%	71.4%	71.9%	178	66	152	380
		Leucocytes +3	-	27.9%	95.5%	82.1%	64.2%	66.8%	92	20	238	426
		Leucocytes +4	-	14.5%	98.2%	85.7%	60.8%	62.6%	48	8	282	438
		Nitrite pos	0.639 (0.613–0.664)	30.3%	97.5%	90.1%	65.4%	68.9%	100	11	230	435
		Leucocytes +1 or nitrite pos	0.782 (0.757–0.823)	74.5%	71.7%	66.1%	79.2%	72.9%	246	126	84	320
		Leucocytes +1 and nitrite pos	-	27.0%	98.9%	94.7%	64.7%	68.3%	89	5	241	441
		Leucocytes +2 and nitrite pos	-	22%	100%	97%	63%	66%	72	2	258	444
		Leucocytes +3 and nitrite pos	-	13%	100%	98%	61%	63%	43	1	287	445
		Leucocytes +4 and nitrite pos	-	6.4%	99.8%	95.5%	59.0%	60.1%	21	1	309	445
UTI	UFC	BACT/µL–7.9/µL	0.743 (0.693–0.793)	95.8%	23.9%	27.2%	95.0%	40.3%	113	303	5	95
*n* = 966	*n* = 512	WBC/µL–15/µL	0.856 (0.819–0.894)	94.9%	48.7%	35.4%	97.0%	59.3%	112	204	6	194
		WBC/µL–448/µL	-	50.8%	91.5%	63.8%	86.3%	82.2%	60	34	58	364
		WBC/µL–1125/µL	-	33.9%	95.0%	66.7%	82.9%	81.0%	40	20	78	378
		BACT/µL–7.9/µL or WBC/µL–15/µl	0.832 (0.793–0.871)	98.3%	21.4%	27.0%	97.7%	39.0%	116	313	2	85
		BACT/µL–7.9/µL and WBC/µL–15/µL	-	92.4%	51.3%	36.0%	95.8%	60.7%	109	194	9	204
	UDA	Leucocytes +1	0.796 (0.0761–0.831)	84.7%	67.1%	43.8%	93.5%	71.2%	160	205	29	418
	*n* = 812	Leucocytes +2	-	67.7%	80.1%	50.8%	89.1%	77.2%	128	124	61	499
		Leucocytes +3	-	34.9%	92.1%	57.4%	82.4%	78.8%	66	49	123	574
		Leucocytes +4	-	19.0%	96.8%	64.3%	79.8%	78.7%	36	20	153	603
		Nitrite pos	0.616 (0.581–0.652)	32.3%	91.0%	52.1%	81.6%	77.3%	61	56	128	567
		Leucocytes +1 or nitrite pos	0.805 (0.771–0.840)	86.2%	64.7%	42.6%	93.9%	69.7%	163	220	26	403
		Leucocytes +1 and nitrite pos	-	30.7%	93.4%	58.6%	81.6%	78.8%	58	41	131	582
		Leucocytes +2 and nitrite pos	-	25%	95%	62%	81%	79%	48	30	141	593
		Leucocytes +3 and nitrite pos	-	17%	98%	71%	80%	79%	32	13	157	610
		Leucocytes +4 and nitrite pos	-	7.9%	98.9%	68.2%	78.0%	77.7%	15	7	174	616

AUROC, Area under receiver operation characteristics curve; BACT/µL, bacterial count per µL; DA, diagnostic accuracy; FN, false negative; FP, false positive; NPV, negative predictive value; PPV, positive predictive value; TN, true negative; TP, true positive; UDA, urinary dipstick analysis; UFC, urine flow cytometry; UTI, urinary tract infection; WBC/µL, white blood cell count per µL.

**Table 3 diagnostics-14-00412-t003:** Model AUROC, diagnostic values, and cross tabulations for each test and selected cut-offs stratified by reference test and subgroup.

Reference Test	Index Test	Subgroup	Variables and Cut-Off	Model AUROC (95%CI)	Sensitivity	Specificity	PPV	NPV	DA	TP	FP	FN	TN
Bacteriuria	UFC	Men	BACT/µL–7/µL or WBC/µL–3.2/µL	0.900 (0.862–0.938)	98.2%	30.0%	48.9%	96.0%	57.6%	107	112	2	48
		*n* = 269	BACT/µL–7/µL and WBC/µL–3.2/µL	-	91.7%	53.1%	57.1%	90.4%	68.8%	100	75	9	85
		Women	BACT/µL–7/µL or WBC/µL–3.2/µL	0.742 (0.677–0.806)	99.0%	3.6%	43.5%	83.3%	44.4%	103	134	1	5
		*n* = 243	BACT/µL–7/µL and WBC/µL–3.2/µL	-	93.3%	18.0%	46.0%	78.1%	50.2%	97	114	7	25
	UDA	Men	Leucocytes +1 or nitrite pos	0.828 (0.789–0.867)	77.7%	79.8%	74.3%	82.6%	78.9%	139	48	40	190
		*n* = 417											
		Women	Leucocytes +1 or nitrite pos	0.731 (0.679–0.782)	70.9%	62.5%	57.8%	74.7%	66.0%	107	78	44	130
		* n * = 359											
UTI	UFC	Men	WBC/µL–15/µl	0.891 (0.849–0.933)	95.7%	61.5%	46.5%	97.6%	70.4%	67	77	3	123
		*n* = 270	WBC/µL–448/µL	-	60.0%	91.5%	71.2%	86.7%	83.3%	42	17	28	183
			WBC/µL–1125/µL	-	42.9%	94.0%	71.4%	82.5%	80.7%	30	12	40	188
		Women	WBC/µL–15/µL	0.798 (0.730–0.866)	93.8%	35.9%	26.2%	95.9%	47.2%	45	127	3	71
		*n* = 246	WBC/µL–448/µL	-	37.5%	91.4%	51.4%	85.8%	80.9%	18	17	30	181
			WBC/µL–1125/µL	-	20.8%	96.0%	55.6%	83.3%	81.3%	10	8	38	190
	UDA	Men	Leucocytes +1 or nitrite pos	0.841 (0.799–0.882)	89.2%	71.8%	52.1%	95.1%	76.3%	99	91	12	232
		*n* = 434	Leucocytes +2 and nitrite pos	-	22.5%	95.4%	62.5%	78.2%	76.7%	25	15	86	308
			Leucocytes +3 and nitrite pos	-	14.4%	97.2%	64.0%	76.8%	76.0%	16	9	95	314
		Women	Leucocytes +1 or nitrite pos	0.758 (0.697–0.818)	82.1%	57.0%	33.2%	92.4%	62.2%	64	129	14	171
		*n* = 378	Leucocytes +2 and nitrite pos	-	29.5%	95.0%	60.5%	83.8%	81.5%	23	15	55	285
			Leucocytes +4 and nitrite pos	-	20.5%	98.7%	80.0%	82.7%	82.5%	16	4	62	296

AUROC, Area under receiver operation characteristics curve; BACT/µL, bacterial count per µL; DA, diagnostic accuracy; FN, false negative; FP, false positive; NPV, negative predictive value; PPV, positive predictive value; TN, true negative; TP, true positive; UDA, urinary dipstick analysis; UFC, urine flow cytometry; UTI, urinary tract infection; WBC/µL, white blood cell count per µL.

## Data Availability

Data cannot be made available due to Danish data protection laws. The full study protocol is published and freely available [11].

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
