# Peer review of "Urine Flow Cytometry and Dipstick Analysis in Diagnosing Bacteriuria and Urinary Tract Infections among Adults in the Emergency Department—A Diagnostic Accuracy Trial"

_diagnostics, 2024, doi:10.3390/diagnostics14040412_

Round 1

Reviewer 1 Report

Comments and Suggestions for Authors

The presented study is to investigate the adequacy of UFC and UDA for the diagnosis of UTI by excluding bacteriuria in the emergency department setting. It is also aimed to determine the cut-off values. The study generally focuses on clinical tuning of the performance of a commercial diagnostic device. The results are analysed neutrally and the data are presented in an understanding manner. minor revision required.

Only Figure 1 is confusing. The phrasing is problematic (No=number or No=absent, etc). Numbers in the figure contradict the following paragraph (e.g. how many urine cultures were performed? 776 in the figure, 786 in the text). for "reference test UTI diagnosis" n=?????? etc.. Flowchart should be made clearer and contradictions with the text should be corrected. 

---

Comments on the Quality of English Language

None

Author Response

Thank you for your time and valuable comments.

Regarding your feedback on figure one about the meaning of “no”, you are absolutely correct. It is ambiguous and easily corrected to a more precise phrasing of “UFC not done” instead. Thank you for spotting this improvement.

The apparent contradiction of numbers between text and figure is due to the STARD standard of putting the index test first and the reference test last. So the 786 in the text is out of the total 966 patients included in the trial, whereas the 776 in the UDA part of the is out of the 812 patients who had UDA done, and the 512 in the UFC part of the figure is out of the 516 who had UFC done.

We have added a sentence to the results text in the section about the 786 (lines 191-194) clarifying this. “Out of 516 patients where UFC was performed, 512 also had urine cultures done, while out of 812 patients who had UDA performed, 776 had urine cultures done.”

The missing “reference test UTI diagnosis" is also easily amendable, since the box just had to be one made pixel taller for it to appear, and I am just ashamed that this has not been noticed in my many read-throughs.

Thank you again for your helpful and relevant feedback.

Reviewer 2 Report

Comments and Suggestions for Authors

Thank you for giving me the opportunity to review this work.

Your introduction begins with describing a couple of studies, which is a bit unusual as this is usually done in the discussion chapter.

Your aim is to find out if the tests you evaluate are able to identify infection. Now it is obvious that cultures offer a lot more information, with the "price" of a longer wait time. I suggest you elaborate on this aspect, otherwise we already know the answer to your problem.

An important aspect worth discussing is the price of the tests in comparison with the price of the culture. We know that the culture takes longer than the other tests, it would be interesting to know if the total price goes up or down.

Another aspect I would discuss is if we can prescribe antibiotics in the ER based on dipstick or flowcitometry - maybe even legal aspects.

I will gladly revise an updated version of your paper.

Author Response

Thank you for your time and valuable comments.

The section in the introduction where the studies are discussed (lines 48-55) arose from the need to express that while very good at detecting bacteriuria, urine culture performance suffers when used to diagnose urinary tract infections. We changed the wording of that section to a more concise statement of these facts with the same references.

"Although urine cultures are highly sensitive and specific for bacteriuria, their performance in diagnosing UTIs suffers. Sensitivity can be as low as 45%, specificity as low as 72%, and negative predictive value (NPV) as low as 50% [5-7]."

It states the same, but without the explanation or discussion, which you are right in preferring to be in the discussion section.

Regarding the turnaround time of urine cultures and their impact on diagnostics in the ED, we have a sentence covering this in lines 57-59. However, we agree that more space is needed to underline this point. We added the following to the discussion after line 320, where it fits well and underlines the point.

"Urine cultures are invaluable in determining the aetiology and resistance patterns of UTIs and thus provide vital information for the treatment of UTIs. However, poor diagnostic performance for UTIs and the fact that they take 24 to 48 hours makes them unsuitable for diagnosing UTIs in the ED. Thus, faster and better diagnostic tools are needed in the ED."

Your point about price is very acute. If UFC or UDA is used to rule out the need for urine cultures, it would be helpful to know if money is saved when doing so. However, it is not easily calculated since it is hard to determine the cost of each test, since they differ depending on what setup is used, how many tests are performed and what the local cost of urine cultures is. While very relevant, this cost-benefit analysis would be a paper in itself and beyond the scope of this article.

Although the main motive for the primary author's interest in UTI diagnostics is antibiotics and antibiotic prescription, it is tough to advise from the results we have. Since both UFC and UDA lack specificity and PPV, we cannot advise using them to diagnose UTIs, only to rule them out. As our population consists of patients admitted with suspicion of infection, ruling out UTIs does not rule out the need for antibiotics. Only in very select situations where another infection is certain, but there is doubt whether the patient has a UTI as well, could it impact the choice of antibiotics. Furthermore, it would depend on local prescription guidelines and resistance patterns and thus be hard to make generalisable recommendations.

Again, thank you for your feedback. Your insights and suggestions are invaluable for improving our paper, and we value them highly.

Round 2

Reviewer 2 Report

Comments and Suggestions for Authors

Thank you for making the changes I suggested.